# Functional Analysis of Forkhead Transcription Factor Fd59a in the Spermatogenesis of *Drosophila melanogaster*

**DOI:** 10.3390/insects15070480

**Published:** 2024-06-27

**Authors:** Ting Tang, Mengyuan Pei, Yanhong Xiao, Yingshan Deng, Yuzhen Lu, Xiao-Qiang Yu, Liang Wen, Qihao Hu

**Affiliations:** 1Guangdong Provincial Key Laboratory of Insect Developmental Biology and Applied Technology, Guangzhou Key Laboratory of Insect Development Regulation and Application Research, Institute of Insect Science and Technology, School of Life Sciences, South China Normal University, Guangzhou 510631, China; 15105987674@163.com (T.T.); 2022022993@m.scnu.edu.cn (M.P.); 2020022814@m.scnu.edu.cn (Y.X.); ydengys@163.com (Y.D.); luyuzhen@m.scnu.edu.cn (Y.L.); xqyu@m.scnu.edu.cn (X.-Q.Y.); 2National Key Laboratory of Green Pesticide, College of Plant Protection, South China Agricultural University, Guangzhou 510642, China

**Keywords:** spermatogenesis, Forkhead, FoxD, testis, apoptosis

## Abstract

**Simple Summary:**

Spermatogenesis, which is regulated by many different genes, is a conserved process across species to produce mature sperm for animal reproduction. Fox transcription factors can bind to DNA sequences in the promoters to regulate gene expression. FoxD subfamily members are mainly involved in metabolism and early organ development. In *Drosophila melanogaster*, FoxD subfamily member Fd59a may regulate the development of the nervous system and control the egg-laying behavior of females. However, the functions of insect FoxD members are still largely unknown. In this study, we investigated the role of Fd59a in the spermatogenesis of *Drosophila*. We found that mutations in *Fd59a* caused swelling of the apical region in the testis, resulting in fewer mature sperm in the seminal vesicle and significantly lower fertility of *Fd59a* mutant males compared to the control flies. We also found that the homeostasis of the testis stem cell niche in *Fd59a* mutant and RNAi flies was disrupted, causing increased apoptosis of sperm bundles. RNA sequencing and qRT-PCR results suggested that Fd59a can regulate the expression of genes related to reproductive process and cell death. Our collective results indicated that Fd59a plays a key role in *Drosophila* spermatogenesis, which will help to understand the role of FoxD members in insect spermatogenesis.

**Abstract:**

Spermatogenesis is critical for insect reproduction and is regulated by many different genes. In this study, we found that Forkhead transcription factor Fd59a functions as a key factor in the spermatogenesis of *Drosophila melanogaster*. Fd59a contains a conversed Forkhead domain, and it is clustered to the FoxD subfamily with other FoxD members from some insect and vertebrate species. Mutations in *Fd59a* caused swelling in the apical region of the testis. More importantly, fewer mature sperm were present in the seminal vesicle of *Fd59a* mutant flies compared to the control flies, and the fertility of *Fd59a^2/2^* mutant males was significantly lower than that of the control flies. Immunofluorescence staining showed that the homeostasis of the testis stem cell niche in *Fd59a^2/2^* mutant and *Fd59a* RNAi flies was disrupted and the apoptosis of sperm bundles was increased. Furthermore, results from RNA sequencing and qRT-PCR suggested that Fd59a can regulate the expression of genes related to reproductive process and cell death. Taken together, our results indicated that Fd59a plays a key role in the spermatogenesis of *Drosophila*.

## 1. Introduction

The Forkhead box (Fox) transcription factor, which contains a highly conserved DNA binding domain of ~100 amino acids consisting of three α helices, three β folds, and two ring connections, plays critical roles in organ development, innate immunity, and other processes [1]. Based on phylogenetic analysis, Fox proteins are assigned to different subclasses and named “Fox, subclass N, member X” [2]. FoxD subfamily members are mainly involved in metabolism and early organ development [3]. In mammals, FoxD1 regulates human early embryonic development and is associated with various diseases. For example, FoxD1 can promote *SLC2A1* (Solute carrier family 2 member 1) transcription and inhibit the degradation of *SLC2A1* to facilitate the proliferation, invasion, and metastasis of pancreatic cancer cells [4,5,6]. In planarian, *FoxD* gene expression was induced by wound signaling, and it was involved in head regeneration [7]. In *Drosophila melanogaster*, Fd59a/FoxD may regulate the development of the nervous system and control the egg-laying behavior of females [8]. However, the functions of insect FoxD subfamily members are still largely unknown.

Spermatogenesis is a process to produce mature sperm for reproduction. The process of spermatogenesis is conserved from insects to vertebrates; thus, insect testis is an ideal model for studying the mechanisms of spermatogenesis [9]. In *D. melanogaster*, germ stem cells (GSCs) differentiate into goniablasts under the control of the stem cell niche; then, goniablasts develop into spermatids through mitosis and meiosis. After nuclear elongation and individualization processes, round spermatids finally become mature sperm [10]. 

Spermatogenesis is regulated by multiple signaling pathways, such as TGF-β (Transforming Growth Factor β), Notch, JAK-STAT (Janus kinase-signal transducer and activator of transcription), BMP (Bone morphogenetic protein), and Hedgehog (Hh) pathways [11], and by many genes [12,13]. Recent studies showed that different genes are involved in the spermatogenesis of insects. For example, knockdown expression of *ribosomal protein S3* (*RpS3*) strongly disrupted spermatid elongation and individualization processes in *D. melanogaster* [14]. The knockdown or mutation of the *cytochrome c1-like* (*cyt-c1L*) gene in early germ cells resulted in male sterility of *D. melanogaster* [15]. Moreover, *BmHen1*, a gene in *Bombyx mori* encoding methyltransferase that modifies piRNAs, was found to regulate eupyrene sperm development [16]. These results indicate that the molecular mechanism of insect spermatogenesis is much more complicated than what we have already known about.

In our previous study, we showed that *B. mori* FoxA participated in the development of wing disc [17]. Microarray data showed that Fox genes were expressed in *B. mori* testis, and *BmFoxD* was expressed at a high level [18]. In *Drosophila*, the expression of *Fd59a/FoxD* was also about 2-fold higher in the testis than in the ovary [19], suggesting that Fd59a may play a role in testis development or spermatogenesis. In this study, we found that the mutation and knockdown expression of *Fd59a* caused swelling in the apical region of the testis and decreased male fertility. More importantly, the loss of function of *Fd59a* disrupted the homeostasis of the testis stem cell niche and induced the apoptosis of sperm bundles, resulting in fewer mature sperm in the seminal vesicle. By analyzing RNA sequencing from the testis of *Fd59a^2/2^* mutants, we found that Fd59a may regulate the expression of genes related to reproductive and metabolic processes. Our findings suggest that Fd59a plays a role in *Drosophila* spermatogenesis.

## 2. Materials and Methods

### 2.1. Fly Lines

The wild-type *w^1118^* line was maintained in the laboratory [20]. *Nos*-*Gal4* (TB00040) and *UAS-GFP dsRNA* (BDSC9331) fly lines were obtained from Tsinghua Fly Center in Beijing, China. *Fd59a^1^*/CyO (BDSC56819), *Fd59a^2^*/CyO (BDSC56820), *Df(2R)BSC864* (BDSC29987), and *UAS-Fd59a* RNAi (BDSC31937) flies were purchased from the Bloomington *Drosophila* Stock Center (BDSC) in Indiana, USA. The *Bam-Gal4* fly line was a gift from the laboratory of Professor Yufeng Wang at the School of Life Science, Central China Normal University, Wuhan, China.

To analyze the functions of Fd59a, *Fd59a^1^/CyO* males were crossed with *Fd59a^1^/CyO* females to generate *Fd59a^1/1^* loss-of-function flies, while *Fd59a^2^/CyO* males were crossed with *Fd59a^2^/CyO* females to generate *Fd59a^2/2^* loss-of-function flies. To knock down the expression of *Fd59a*, *Nos-Gal4* and *Bam-Gal4* flies were crossed with *UAS-Fd59a* RNAi flies.

All flies were reared on a fresh cornmeal/yeast/brown sugar diet with p-hydroxybenzoic acid methylester as a mold inhibitor at 25 °C with a photoperiod of approximately 12 L/12 D (light/dark) [21]. 

### 2.2. Bioinformatics Analysis

The amino acid sequences of Fd59a and its homologous proteins were obtained using protein BLAST at the National Center for Biotechnology Information (NCBI, https://blast.ncbi.nlm.nih.gov, accessed on 29 February 2024). Sequence alignment was performed by Cluster W. The construction of a phylogenetic tree was achieved by using RAxML (Random Axelerated Maximum Likelihood), approached with 1000 bootstrap replications [22]. The identification of functional protein domains in Fd59a and its homologous proteins was performed by SMART (https://smart.embl.de/, accessed on 29 February 2024). The prediction of potential Fox binding sites in the promoter sequences of selected genes was accomplished using the JASPAR program (https://jaspar.genereg.net/, accessed on 29 February 2024). 

### 2.3. RNA Isolation and Quantitative RT-PCR

To investigate the expression profile of the *Fd59a* gene from the embryo to adult stages, approximately 50 adult *w^1118^* flies (male/female ratio ~2:1) were collected in a cage for mating. Then, the flies were relocated to a fresh cage at 2 h intervals. Embryos at 2, 4, 6, 8, 10, 12, 14, 16, 18, 20, 22, and 24 h after egg laying; the 1st, 2nd, and 3rd instar larvae; pupae at the early, middle, and late stages; and 1-, 3-, and 5-day-old adults were collected. All samples were collected in RNAex Pro reagent (1000 µL) (Accurate Biology, Changsha, China) and stored in a −80 °C freezer for subsequent RNA isolation.

Total RNA was isolated from the above samples using a method described previously [20]. The first-strand complementary DNA (cDNA) was synthesized from 1 μg of total RNA using the HiFiScript gDNA removal cDNA Synthesis Kit (cwbiotech, Taizhou, China). To analyze the expression of target genes, gene-specific primers (Table 1) were designed based on the sequences available in the Flybase. All primers were synthesized by Tsingke Biotechnology (Beijing, China). 

Quantitative reverse transcription—polymerase chain reaction (qRT-PCR) was conducted by using QuantStudio™ 6 Flex (Thermo Fisher Scientific, Waltham, MA, USA) and ChamQ SYBR qPCR Master Mix (Vazyme, Nanjing, China), following the manufacturer’s instructions. The qPCR cycling program was set as 95 °C for 30 s, succeeded by 40 cycles of 95 °C for 10 s and 60 °C for 30 s. Relative gene expression was normalized to the endogenous reference gene *Rp49* by using the comparative CT (2^−ΔΔCt^) method [23].

### 2.4. Male Fertility Test

To evaluate the reproductive ability of male *Fd59a* mutant flies, a male fertility test was conducted. Ten 1-day-old virgin *w^1118^* females were housed with fifteen 3-day-old *w^1118^* or *Fd59a^2/2^* males in vials containing egg collection plates to collect eggs. The egg collection plates were replaced every 24 h, and the number of embryos and larvae in the plates was counted as described previously [24].

### 2.5. Immunofluorescence Staining

Testes of 3-day-old adult flies were dissected in 10 mM phosphate-buffered saline (PBS) (137 mM NaCl, 2.7 mM KCl, 10 mM Na_2_HPO_4_, 2 mM KH_2_PO_4_, pH 7.4) and fixed in 4% paraformaldehyde (PFA) prepared in PBS for 40 min at room temperature. Testis samples were washed in 3‰ PBT (10 mM PBS with 0.3% Triton X-100) at least 3 times (each for 15 min) and treated with blocking buffer (3‰ PBT with 5% normal goat serum) for 1 h at room temperature. Then, samples were incubated at 4 °C overnight with primary antibodies diluted in dilution buffer (3‰ PBT with 3% normal goat serum), washed three times in 3‰ PBT to remove unbound antibodies, and subsequently incubated with secondary antibodies diluted in a dilution buffer at room temperature for 3 h in darkness. Samples were washed at least three times with 3‰ PBT and mounted using VECTASHIELD^®^ antifade Mounting Medium containing 1.5 μg/mL DAPI (Vector Laboratories, Newark, NJ, USA). 

The primary antibodies used in this study were as follows: rat anti-Vasa (1:50, Developmental Studies Hybridoma Bank, Iowa, IA, USA), mouse anti-Fasciclin III (Fas III) (1:100, Developmental Studies Hybridoma Bank, 7G10, Iowa, IA, USA), and mouse anti-αSpectrin (1:50, Developmental Studies Hybridoma Bank, 3A9, Iowa, IA, USA). The secondary antibodies used were Alexa Fluor-conjugated goat anti-mouse 568 and goat anti-rat 488 antibodies (1:500, Invitrogen, Carlsbad, CA, USA). Fluorescent images were captured using an FV3000 confocal microscope (Olympus, Tokyo, Japan).

### 2.6. TUNEL Assay

To analyze whether loss of function of Fd59a could induce cell death in sperm bundles, testes from 3-day-old adults of *Fd59a^2/2^* and *Fd59a* RNAi flies were dissected. The collected testes were fixed in 4% paraformaldehyde and rinsed in 3‰ PBT, as described above.

A terminal deoxyribonucleotide transferase (TdT)-mediated dUTP nick end labeling (TUNEL) assay was conducted using the TUNEL assay kit (C1088 and C1090, Beyotime Biotechnology, Shanghai, China). Testis samples prepared as above were incubated with a TUNEL reaction mixture containing 5 μL of TdT enzyme, 45 μL of fluorescent labeling solution, and a moderate enzyme dilution buffer at 4 °C overnight and then washed three times in 3‰ PBT. DAPI staining was performed as described above.

### 2.7. RNA Sequencing

Testes from 3-day-old *Fd59a^2/2^* and *w^1118^* flies were dissected in DEPC-treated PBS (10 mM, pH7.4). Total RNA was extracted, and RNA sequencing was conducted by Shanghai Majorbio Biotech (Shanghai, China) using Illumina HiSeq 2000 (Illumina, San Diego, CA, USA).

### 2.8. Bioinformatics Analysis of RNA-seq Data

The transcript expression levels were quantified by Fragments Per Kilobase per Million mapped (FPKM) read values, and differentially expressed genes (DEGs) were identified based on a |log_2_ fold-change| > 1.5 along with a *p*-value adjustment below 0.05 across three biological replicates. The gene ontology (GO) enrichment of DEGs was analyzed using the GOseq R package (version 3.9).

### 2.9. Statistical Analysis

The size of the apical region from *Fd59a^2/2^* and *w^1118^* testes was quantitated by ImageJ (version 1.54j). For each sample, three biological replicates and at least three technical replicates for each biological sample were performed. Data were presented as the mean ± S.E. (standard error); significant differences were analyzed by the Student’s *t*-test (for comparison between two groups) or by one-way analysis of variance followed by a least significant difference test (for multiple comparisons among groups) using GraphPad Prism version 9.0.

## 3. Results

### 3.1. Expression Profile of Fd59a in Drosophila

To understand the functions of *Fd59a*, we first determined the expression profile of *Fd59a* at different developmental stages of *Drosophila* by qRT-PCR. The results showed that the expression of *Fd59a* mRNA peaked twice during development from embryo to adult stages, with the first peak around the mid-embryonic stage and then a plateau with a relatively high level from the late embryonic stage to the first instar larval stage and the second peak just at the metamorphosis period, maintaining at a relatively high level until the mid-pupal stage (Figure 1A), indicating that Fd59a may be involved in *Drosophila* development.

It has been reported that the loss of function of Fd59a affected the egg laying of *Drosophila* females [8]. Interestingly, FlyAtlas anatomical expression data from the Flybase showed that the expression level of *Fd59a* was about 2-fold higher in the testis than in the ovary [19]. We performed qRT-PCR and confirmed that mRNA levels of *Fd59a* were significantly higher in the testis than in the ovary of 3-day-old *w^1118^* flies (Figure 1B), suggesting that Fd59a may also have a function in the testis of *Drosophila*.

### 3.2. Sequence and Phylogenetic Analyses of Fd59a

The *Fd59a* gene is in the second chromosome. The full-length cDNA of *Fd59a* is 1371 bp long, encoding a protein of 456 amino acid residues, with a calculated molecular weight of 49.1 kDa and pI of 5.19. 

Fd59a belongs to the FoxD subfamily. To reveal the evolutionary relationship of Fd59a, Fd59a/FoxD homologous sequences from some insect and vertebrate species were blasted and downloaded from NCBI (Table 2), sequence alignment was performed, and a phylogenetic tree was constructed. The results showed that all the selected Fd59a/FoxD homologous proteins contained a conserved Forkhead box domain (Figure 2A). *Drosophila* Fd59a was the closest to FoxD3-A of *Papilio Xuthus* and FoxD3-like of *Blattella germanica* (Figure 2B) and they contained similar functional domains (Figure 2C). These results suggest that *Drosophila* Fd59a is evolutionarily close to insect FoxD members.

### 3.3. Loss of Function of Fd59a Affects Testis Development and Male Fertility

To determine the role of *Fd59a* in testis development in *D. melanogaster*, testes from *Fd59a^1/1^
*and *Fd59a^2/2^* loss-of-function mutant flies as well as *w^1118^* flies were dissected. *Fd59a^1^* and *Fd59a^2^* were two mutant types of *Fd59a*, with *Fd59a^1^* as a hypomorphic mutation and *Fd59a^2^* as a null allele [8]. We found that the apical region of testes from the *Fd59a^1/1^* and *Fd59a^2/2^
*mutant flies was swelled (Figure 3A(a1,b1,c1),B), with fewer mature sperm in the seminal vesicle of *Fd59a^2/2^* mutant testis compared to many mature sperm in the *w^1118^* flies (Figure 3A(a2,a3,b2,b3,c2,c3)). As *Fd59a^2^* was derived from EMS-based genetic screening, it may have contained unmapped mutations in the second chromosome. To exclude the possibility that the observed phenotypes were derived from other mutations in the second chromosome, we generated the hemizygous flies by crossing *Fd59a^2^* flies with *Df(2R) BSC864* flies, which contained a deletion encompassing the *Fd59a* locus, and similar results were observed (Figure 3A(d1–d3,e1–e3)). As the phenotype of *Fa59a^2/2^* flies was more significant than that of *Fa59a^1/1^* flies, we carried out the following studies in the *Fa59a^2/2^* mutant flies. When *Fd59a^2/2^* males were crossed with *w^1118^* females, the hatching rate of F1 flies was significantly decreased compared to the control (Figure 3C). Together, these results suggest that Fd59a plays a critical role in the development of the testis and/or spermatogenesis of *D. melanogaster*.

### 3.4. Loss of Function of Fd59a Affects Spermatogenesis

At the apical region of testis, about 10 hub cells cluster together and are surrounded by GSCs and CySCs to form the stem cell niche, which governs the proliferation and differentiation of GSCs and CySCs. The disruption of niche homeostasis can lead to swelling of the testis and defects in spermatogenesis [25]. To clarify the role of Fd59a in spermatogenesis, Fas III and Vasa antibodies were used to specifically mark the hub cells and germ cells, respectively, and αSpectrin antibody was employed to identify spectrosomes and fusomes, which are crucial for the early development of germ cells. As a result, the distribution of GSCs and CySCs was scattered in the *Fd59a^2/2^* testis compared to the control testis (Figure 4A(a4,b4)), and a strong pattern of spectrosome and fusome formation was displayed in the control testis, while fewer spectrosomes and fusomes were observed in the *Fd59a^2/2^* mutant testis (Figure 4A(a2,a3,b2,b3),B,C). These results suggest that Fd59a may play a role in maintaining the homeostasis of the testis stem cell niche.

Mammalian FoxD1 is related to apoptosis, as the knockdown expression of *FoxD1* facilitates apoptosis in HNSCC (head and neck squamous cell carcinoma) cells [26]. To determine whether the loss of function of Fd59a could induce apoptosis in the testis, a TUNEL assay was performed. TUNEL positive signals were detected in sperm bundles of the *Fd59a^2/2^* mutant flies but not in the *w^1118^* flies (Figure 5A(a1–a4,b1–b4)), suggesting that the loss of function of Fd59a induced the apoptosis of spermatid.

To further confirm the role of Fd59a in the testis, the expression of *Fd59a* in GSCs was knocked down by *Nos-Gal4* (Figure 4D), and similar phenotypes, such as swelling in the apical region of the testis, fewer mature sperm in the seminal vesicle, and the apoptosis of sperm bundles, were observed in the *Nos-Gal4>Fd59a* RNAi flies (Figure 4A(c1–c3,d1–d3),B,C and Figure 5A(c1–c4,d1–d4)). The knockdown expression of *Fd59a* in the 4–16 stages of spermatogonia by *Bam-Gal4* (Figure 5C) also induced the apoptosis of sperm bundles (Figure 5A(e1–e4,f1–f4)). Moreover, only a few mature sperm were observed in the seminal vesicles of the *Nos-Gal4>Fd59a* RNAi and *Bam-Gal4>Fd59a* RNAi flies, while the control flies were filled with mature sperm (Figure 5B). These combined results suggest that the loss of function of Fd59a in the testis resulted in the disruption of the stem cell niche during spermatogenesis and increased the apoptosis of sperm bundles, finally leading to fewer mature sperm in the seminal vesicle.

### 3.5. Fd59a Regulates Gene Expression in the Testis

To further clarify the role of Fd59a in spermatogenesis, RNA sequencing (RNA-seq) was performed with RNAs isolated from the testes of *w^1118^* (control) and *Fd59a^2/2^* males. In total, 1863 differentially expressed genes (DEGs) with at least a 1.5-fold change (*p*-adjust < 0.05) were identified by RNA-seq, with 854 genes upregulated and 1009 genes downregulated in the testis of *Fd59a^2/2^* flies (Figure 6A). This result suggests that Fd59a may function as a transcription factor in the testis.

Gene ontology (GO) analysis revealed that 120 differentially expressed genes (DEGs) are associated with the reproductive process and 475 DEGs engage in the metabolic process (Figure 6B). Within DEGs related to reproduction, several have already been reported to contribute to gonad development and spermatogenesis, such as *Fz2* and *Zpg* [27,28]. In addition, 57 DEGs were implicated in cell death, including *Rnrs* and *Ptp52F* [29,30].

To confirm the RNA-seq data, 20 DEGs associated with reproductive process and cell death were chosen for qRT-PCR validation (Table 3). The expression patterns of these DEGs in the testis were consistent with those of the RNA-seq data (Figure 6C). Then, 2000 bp promoter sequences upstream of the transcriptional start sites of these selected genes were downloaded, and the potential Fox binding sites were predicted using the JASPAR database (the relative profile score threshold was set to 85%). Except for the *CG32817* promoter, several Fox binding sites were predicted in the promoter of each selected gene, with more than 10 potential Fox binding sites in the promoters of the *Spd-2*, *Cal1*, *Blanks*, *Ptp52F*, *Lola*, and *Debcl* genes (Table 3). This result further supports that Fd59a is an upstream regulator of these genes. Taken together, our results suggest that Fd59a serves as a transcription factor to regulate the expression of genes involved in reproduction, cell growth, and cell death in the testis directly or indirectly.

## 4. Discussion

*Drosophila* testis is an ideal system for studying spermatogenesis. In this study, we found that FoxD transcription factor Fd59a contributes to the spermatogenesis of *Drosophila*. So far, little is known about the functions of *Drosophila* Fd59a/FoxD and other insect FoxD members. It was reported that the loss of function of Fd59a affects the female egg-laying behavior of *Drosophila* [8]. Moreover, *Drosophila* CHES-1-like/FoxN suppressed the differentiation of germline stem cells by upregulating *Dpp* expression, whereas ectopic expression of *CHES-1-like* led to a significant decrease in male fertility [31]. In *B. mori*, Fox family genes were expressed in the testis, with *BmFoxL2-2* and *BmFoxD* at a higher level than other BmFox genes [18]. However, the function of BmFoxD in the testis is still unknown. 

In this study, we showed that spermatogenesis was disrupted and the apoptosis of sperm bundles was induced in *Fd59a* mutant and RNAi flies. Spermatogenesis is a complex process regulated by multiple signaling pathways and many different genes. The over-activation of the JAK-STAT signaling pathway led to the overgrowth of the testis and the disrupted structure of the testis stem cell niche in *Drosophila* [25,32]. The over-activation of the JNK or loss of the Notch signaling caused cell death in the testis of *Drosophila* [33,34]. In mammals, the deletion of *Stat3* in the Foxd1 cell lineage protected mice from kidney fibrosis [35]. Hypoxia-inducible factors (HIFs) regulated genes related to oxygen homeostasis, and a lack of Hif-p4h-2 (HIF prolyl-4-hydroxylases) in the FoxD1 lineage led to the dysregulation of genes involved in the Notch signaling pathway [36]. These results suggest that there is a genetic interaction between mammalian FoxD subfamily members and the JAK-STAT and Notch pathways. However, mRNA levels of genes related to the above three signaling pathways did not change significantly in the *Fd59a^2/2^* testis (RNA-seq data), indicating that Fd59a is not involved in the JAK-STAT, JNK, or Notch signaling pathway. Therefore, insect FoxD members may function differently from mammalian FoxD subfamily members.

It has been shown that Fd59a is expressed in octopaminergic neurons and that it regulates the egg-laying behavior of female *Drosophila* [8]. In the Chinese mitten crab (*Eriocheir sinensis*), the expression of the octopamine receptor changed significantly in the androgenic gland (AG) between the proliferation and secretion phases [37]. We showed that the loss of function of Fd59a caused defects in spermatogenesis. These combined results suggest that octopaminergic neurons and octopamine may play a role in spermatogenesis, which needs to be determined by further study.

The results from RNA-seq and qRT-PCR showed that many genes related to reproduction and cell death were differentially expressed in the testis of *Fd59a^2/2^* flies. Among the reproduction-related DEGs, *Fz2* and *Zpg* have been reported to regulate germ stem cell development in *Drosophila* testis [27,28], while *Blanks* functioned in sperm individualization [38]. Among the cell death-related DEGs, *Ptp52F* enhanced autophagy and apoptosis in the *Drosophila* midgut [30]. In addition, several potential Fox binding sites were predicted in the promoters of selected DEGs. Thus, Fd59a acts as a transcription factor to regulate the expression of genes involved in spermatogenesis and maintain the survival of sperm cells. 

It was reported that octopamine was essential for increasing GSCs in mating *Drosophila* females [39], and the β-adrenergic-like octopamine receptor (OctβR) was strongly expressed in adult testis [40]. In *Fd59a^2/2^* adult testis, *Octβ2R* expression was downregulated; thus, it is possible that Fd59a regulates spermatogenesis partly through regulating the expression of *Octβ2R*, and *Fd59a* may be a key factor linking the nervous system to the male reproduction system.

## Figures and Tables

**Figure 1 insects-15-00480-f001:**
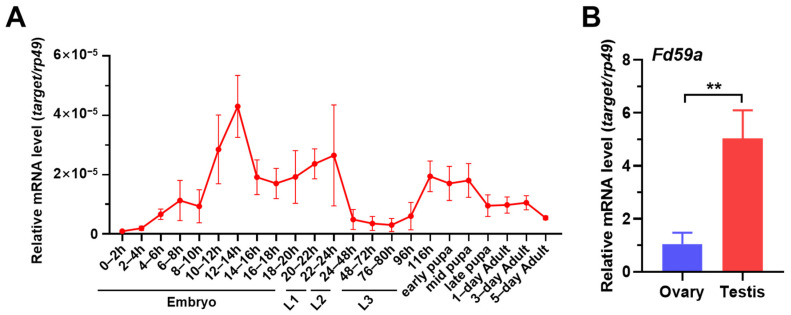
Expression of *Fd59a* at different developmental stages and in adult ovary and testis. (**A**) Expression of *Fd59a* at different developmental stages. *Drosophila* embryos at 2, 4, 6, 8, 10, 12, 14, 16, 18, 20, 22, and 24 h after egg laying; first (L1), second (L2), and third (L3) instar larvae; early, middle, and late pupae; and 1-, 3-, and 5-day-old adult flies were collected to prepare total RNAs for the analysis of the transcriptional expression of *Fd59a* by qRT-PCR. (**B**) Expression of *Fd59a* in the testis and ovary of 3-day-old adult flies. Data were presented as means ± S.E., and significant differences were determined by the Student’s *t*-test and indicated by asterisks. ** *p* < 0.01.

**Figure 2 insects-15-00480-f002:**
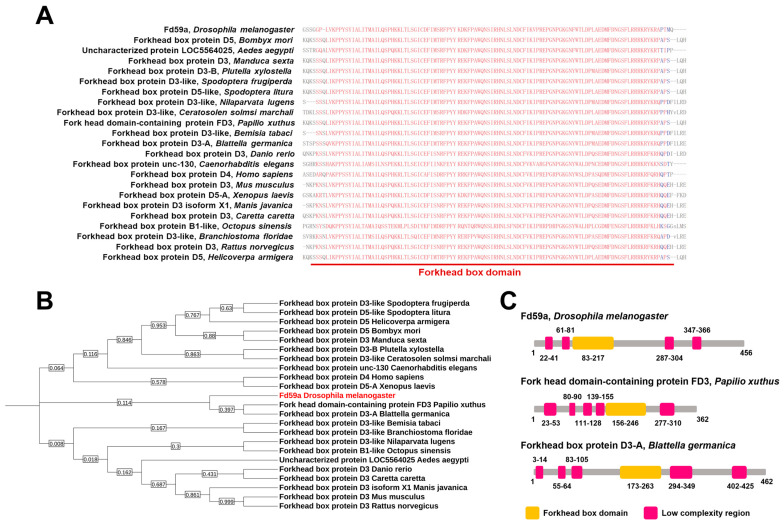
Sequence analysis of *D. melanogaster* Fd59a. Sequence alignment (**A**), Maximum Likelihood phylogenetic tree (**B**), and functional domains (**C**) of Fd59a with its homologous proteins from some insect and vertebrate species. For detailed information on the sequences, see Table 2.

**Figure 3 insects-15-00480-f003:**
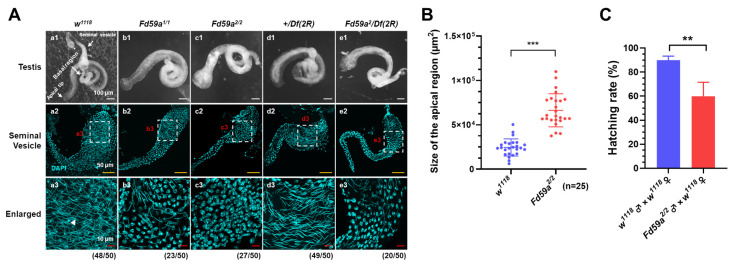
Mutations in *Fd59a* affect the testis development and reduce the number of mature sperm in the seminal vesicle. (**A**) Overall appearance of testes and seminal vesicles from *w^1118^*, *Fd59a^1/1^*, *Fd59a^2/2^*, *+/Df(2R)*, and *Fd59a^2^/Df(2R)* flies. Nuclei were stained with DAPI. Scale bar is 100 µm in (**a1**,**b1**,**c1**,**d1**,**e1**), 50 µm in (**a2**,**b2**,**c2**,**d2**,**e2**), and 10 µm in (**a3**,**b3**,**c3**,**d3**,**e3**). The arrowhead in (**a3**) represents mature sperm. Numbers below the images indicate the pairs of testes with similar phenotypes in the images. (**B**) Quantitative measurements of the apical region of testis from *Fd59a^2/2^* and *w^1118^
*flies. (**C**) Fertility test of *w^1118^* and *Fd59a^2/2^
*male flies. Data are presented as means ± S.E. Significant differences were determined by the Student’s *t*-test and are indicated by asterisks ** *p* < 0.01, *** *p* < 0.001.

**Figure 4 insects-15-00480-f004:**
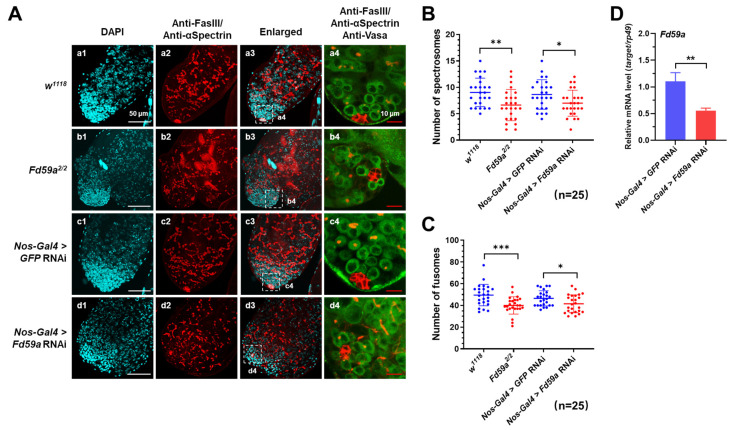
Loss of function of Fd59a in the testis disrupts the homeostasis of the testis stem cell niche. (**A**) Immunostaining of testis. Testes from the *w^1118^*, *Fd59a^2/2^*, *Nos-Gal4>GFP* RNAi, and *Nos-Gal4>Fd59a* RNAi flies were labeled with anti-Vasa (green), anti-FasIII (red), and anti-αSpectrin (red) antibodies, and nuclei were stained with DAPI (blue). (**a1**–**a3**,**b1**–**b3**,**c1**–**c3**,**d1**–**d3**) The apical region of the testis and (**a4**,**b4**,**c4**,**d4**) the enlarged part of the apical tip showing the stem cell niche. In the testis of *Fd59a^2/2^* and *Nos-Gal4>Fd59a* RNAi flies, the distribution of germ cells labeled with anti-Vasa (green) antibody was disrupted, and fewer fusomes and spectrosomes labeled with anti-αSpectrin (red) antibody were observed. Scale bar is 50 µm in (**a1**–**a3**,**b1**–**b3**,**c1**–**c3**,**d1**–**d3**) and 10 µm in (**a4**,**b4**,**c4**,**d4**). (**B**,**C**) The numbers of spectrosomes (**B**) and fusomes (**C**) in the testes of *w^1118^*, *Fd59a^2/2^*, *Nos-Gal4>GFP* RNAi, and *Nos-Gal4>Fd59a* flies. (**D**) Expression of *Fd59a* in the testis of *Nos-Gal4* RNAi flies. Data are presented as means ± S.E. Significant differences were determined by the Student’s *t*-test and are indicated by asterisks. * *p* < 0.05, ** *p* < 0.01, and *** *p* < 0.001.

**Figure 5 insects-15-00480-f005:**
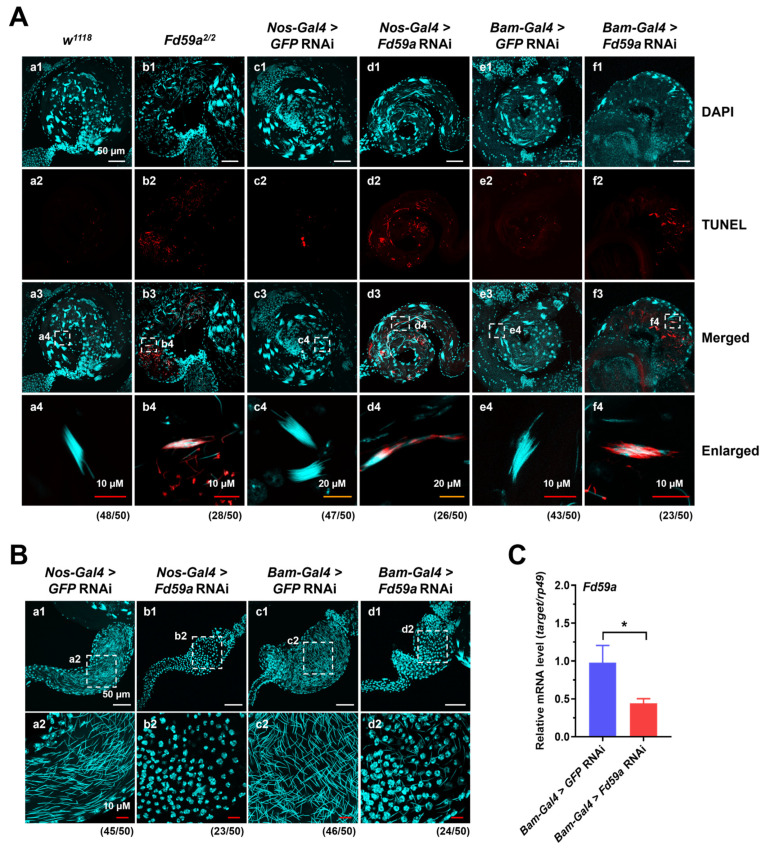
Loss of function of Fd59a in the testis induces apoptosis of sperm bundles. (**A**) Detection of apoptotic cells in the testis. Testes from the *w^1118^*, *Fd59a^2/2^*, *Nos-Gal4>GFP* RNAi, *Nos-Gal4>Fd59a* RNAi, *Bam-Gal4>GFP* RNAi, and *Bam-Gal4>Fd59a* RNAi flies were stained with TUNEL assay for apoptotic cells (red), and nuclei were stained with DAPI (blue). (**a1**–**a3**,**b1**–**b3**,**c1**–**c3**,**d1**–**d3**,**e1**–**e3**,**f1**–**f3**) The basal region of testis and (**a4**,**b4**,**c4**,**d4**,**e4**,**f4**) the enlarged part of the basal region showing the sperm bundles. In the testis of *Fd59a^2/2^*, *Nos-Gal4>Fd59a* RNAi, and *Bam-Gal4>Fd59a* RNAi flies, many TUNEL signals (red) were detected in the sperm bundles; only a few TUNEL signals were detected in the basal region of *w^1118^*, *Nos-Gal4>GFP* RNAi, and *Bam-Gal4>GFP* RNAi flies, but not in the sperm bundles. Scale bar is 50 µm in (**a1**–**a3**,**b1**–**b3**,**c1**–**c3**,**d1**–**d3**,**e1**–**e3**,**f1**–**f3**), 10 µm in (**a4**,**b4**,**e4**,**f4**), and 20 µm in (**c4**,**d4**). (**B**) DAPI staining of seminal vesicle. Seminal vesicles from *Nos-Gal4>GFP* RNAi, *Nos-Gal4>Fd59a* RNAi, *Bam-Gal4>GFP* RNAi, and *Bam-Gal4>Fd59a* RNAi flies were stained with DAPI. Scale bar is 50 µm in (**a1**,**b1**,**c1**,**d1**) and 10 µm in (**a2**,**b2**,**c2**,**d2**). Numbers below the images indicate the pairs of testes with similar phenotypes in the images. (**C**) Expression of *Fd59a* in the testis of *Bam-Gal4* RNAi flies. Data are presented as means ± S.E. Significant differences were determined by the Student’s *t*-test and are indicated by asterisks. * *p* < 0.05.

**Figure 6 insects-15-00480-f006:**
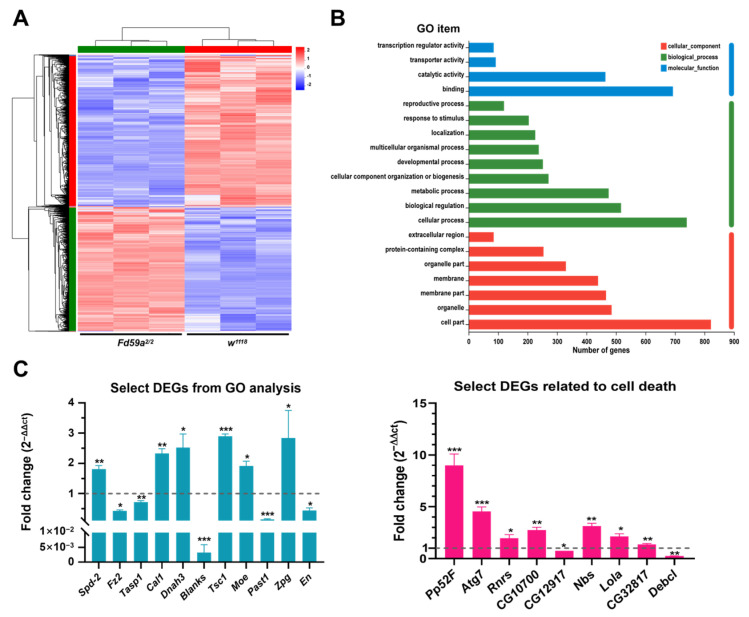
RNA-seq analysis of RNAs from the testes of *Fd59a^2/2^* and *w^1118^* flies. (**A**) Heatmap of differentially expressed genes (DEGs) in the testis of *Fd59a^2/2^* flies relative to *w^1118^* flies. (**B**) GO analysis of DEGs in the testes between *Fd59a^2/2^* and *w^1118^* flies. (**C**) qRT-PCR validation of selected DEGs from RNA sequencing data. Data are presented as means ± S.E. Significant differences were determined by the Student’s *t*-test and are indicated by asterisks * *p* < 0.05, ** *p* < 0.01, *** *p* < 0.001.

**Table 1 insects-15-00480-t001:** Primers used in this study.

Name	Forward Primer (5′–3′)	Reverse Primer (5′–3′)
*Fd59a*-qRT	CAGGGAAGTCAGTCGGGGGA	GTCGCCACATCGAAGGCGTA
*Rp49*-qRT	GCCCAAGGGTATCGACAACA	ACCTCCAGCTCGCGCACGTT
*Spd-2*-qRT	GTGACCCACACGACCCTCTG	GCCGAATGACCAGCCGTTTG
*Fz2*-qRT	TCGCGAGTCACAATTGCACC	GGACGCCACTCTACGGTGTT
*Tasp1*-qRT	CGGCATGCGAGTCTGTTCGG	ACACAAGGCAGCGCAAGTCTA
*Debcl*-qRT	ATCGACAACGGCGGATGGTT	ACGCGATCCCAAGCGAATCT
*Ptp52F*-qRT	TGTCCGACGATCTTTGCGCT	GGCGTAGGGGGAAAGTGGAC
*Atg7*-qRT	TACAACTGCTGGCCGATGAGG	GCACGGAAAGGCGAACCAAT
*RnrS*-qRT	GGACCGTTTGCTCGTGGAGT	GAAATCCGCGTCCAGGGTGA
*CG10700*-qRT	TGTGGAGGCTACGGCCAATC	TCACCACGGCTGTTTCCCAA
*CG12917*-qRT	CAGGGGCTTCCTTCAGTCGG	AAATAGCCAGACACGGGGGC
*Nbs*-qRT	ATTCCCAAAAGCCGCGCAAG	TGGGTCACCTGCCAAATGCT
*Lola*-qRT	CTGCTGAGATATGCGAGCCAGA	GTTCACAATGGCCTCCGCCT
*Cal1*-qRT	GGTGGTGGACGAGGAAACACT	TCCACAGCCTCCTTTGCCAC
*Dnah3*-qRT	AGAGCTGGCAAGAGCGGAAA	ACATTGCGAGACGTGGCACC
*Blanks*-qRT	ACGGGCCAGGAAAGAGCTTG	ACGGCTTCTTTGGCTCGACA
*En*-qRT	CCAACGACGAGAAGCGTCCA	CTCCGCTCGGTCAGATAGCG
*Tsc1*-qRT	GGTTGGCATGACTGGCTCCT	CACGTCCCGGCTGCTTGATA
*CG32817*-qRT	AATCAAGTGTCTAACCCTGAACTGG	GTTGCGCCATCGAAAAGCAT
*Moe*-qRT	GCCTGCGAGAGGTTTGGTTCTT	TCACGTCCTGGTTCATCACCTT
*Past1*-qRT	ACACCCGATCACACAGCCTC	CGCCTGCACTGTGTGGCTAA
*Zpg*-qRT	GGGGCCTATGTGAGCGACAA	CCGCCCTCCCAAATCTTCCA

**Table 2 insects-15-00480-t002:** Fd59a homologous protein sequences used in the phylogenetic tree.

Proteins	Species	Accession Number
Forkhead box protein D5	*Bombyx mori*	XP_004922516.1
Uncharacterized protein LOC5564025	*Aedes aegypti*	XP_001648348.3
Forkhead box protein D3	*Manduca sexta*	XP_030032680.1
Forkhead box protein D3-B	*Plutella xylostella*	XP_037961710.1
Forkhead box protein D3-like	*Spodoptera frugiperda*	XP_035435321.1
Forkhead box protein D5-like	*Spodoptera litura*	XP_022815528.1
Forkhead box protein D3-like	*Nilaparvata lugens*	XP_039277739.1
Forkhead box protein D3-like	*Ceratosolen solmsi marchali*	XP_011504225.1
Forkhead domain-containing protein FD3	*Papilio xuthus*	KPJ03207.1
Forkhead box protein D3-like	*Bemisia tabac*	XP_018914635.1
Forkhead box protein D3-A	*Blattella germanica*	PSN41724.1
Forkhead box protein D5	*Helicoverpa armigera*	XP_021187147.2
Forkhead box protein D3	*Danio rerio*	NP_571365.2
Forkhead box protein unc-130	*Caenorhabditis elegans*	NP_496411.1
Forkhead box protein D4	*Homo sapiens*	NP_997188.2
Forkhead box protein D3	*Mus musculus*	NP_034555.3
Forkhead box protein D5-A	*Xenopus laevis*	NP_001081998.1
Forkhead box protein D3 isoform X1	*Manis javanica*	XP_036880296.1
Forkhead box protein D3	*Caretta caretta*	XP_048718258.1
Forkhead box protein B1-like	*Octopus sinensis*	XP_029653697.1
Forkhead box protein D3-like	*Branchiostoma floridae*	XP_035698942.1
Forkhead box protein D3	*Rattus norvegicus*	NP_542952.1

**Table 3 insects-15-00480-t003:** Differentially expressed genes selected for qRT-PCR validation and the number of predicted Fox binding sites in the 2 kb promoter regions.

Gene Symbol	Log_2_ Fold Difference	Relative Expression	Biological Functions	Forkhead Binding Sites
GO analysis-related genes				
*Spd-2*	1.08	Upregulated	Involved in sperm aster formation	13
*Fz2*	−1.66	Downregulated	Germline stem cell niche homeostasis	6
*Tasp1*	−1.35	Downregulated	Involved in spermatogenesis	7
*Cal1*	0.61	Upregulated	Female meiosis chromosome segregation	10
*Dnah3*	0.70	Upregulated	Involved in sperm competition	6
*Blanks*	−0.72	Downregulated	Involved in sperm individualization	11
*Tsc1*	0.76	Upregulated	Negative regulation of developmental growth	4
*Moe*	0.67	Upregulated	Oocyte anterior/posterior axis specification	10
*Past1*	−2.62	Downregulated	Involved in sperm individualization	3
*Zpg*	1.33	Upregulated	Male germline stem cell population maintenance	5
*En*	−1.40	Downregulated	Involved in gonad development	8
Cell death-related genes				
*Ptp52F*	3.44	Upregulated	Involved in larval midgut cell-programmed cell death	15
*Atg7*	1.60	Upregulated	Involved in autophagy	2
*RnrS*	0.68	Upregulated	Involved in activation of cysteine-type Endopeptidase activity involved in apoptotic process	9
*CG10700*	0.822	Upregulated	Involved in execution phase of apoptosis	9
*CG12917*	−0.62	Downregulated	Involved in apoptotic DNA fragmentation	4
*Nbs*	0.78	Upregulated	Involved in intrinsic apoptotic signaling pathway in response to DNA damage	2
*Lola*	0.74	Upregulated	Involved in nurse cell apoptotic process	14
*CG32817*	0.78	Upregulated	Involved in extrinsic apoptotic signaling pathway	0
*Debcl*	−1.92	Downregulated	Programmed cell death involved in cell development	11

## Data Availability

The data presented in this study are available upon request from the corresponding author.

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
