# Peer review of "Functional Analysis of Forkhead Transcription Factor Fd59a in the Spermatogenesis of Drosophila melanogaster"

_insects, 2024, doi:10.3390/insects15070480_

Round 1
Reviewer 1 Report
Comments and Suggestions for Authors
The manuscript by Ting Tang and coauthors explores the role of the Forkhead transcription factor Fd59a in spermatogenesis of Drosophila melanogaster. The authors found that Fd59a mutant flies have deformed testes that produce very few sperm. They also demonstrate that Fd59a is important for regulation of gene expression in testes affecting the processes of the homeostasis and apoptosis.
The abstract correctly summarizes the findings. The introduction is very clear. The paper is an important step in clarifying the role of Forkhead transcription factor in regulating Drosophila spermatogenesis and towards identifying the target genes. The paper is technically solid, and the conclusions are well supported by the data.
Minor comments:
L74: “In Drosophila, expression of Fd59a/FoxD was about 2-fold higher in the testis than in the ovary.” also
L187: “Interestingly, microarray data from the Flybase showed that expression level 187 of Fd59a was about 2-fold higher in the testis than in the ovary.“
However, microarray data available the Flybase show that from the Fd59a/FoxD has higher expression in ovaries than in testes: http://flybase.org/reports/FBgn0004896.htm
Please clarify.
L315: “Then, 2000 bp promoter sequences upstream the transcriptional start sites of these select genes were downloaded, and the potential Fox binding sites were predicted using JASPAR database (the relative profile score threshold was set to 85%).” These data are not available within the manuscript. Please provide these results.
L321: “Taken together, our results suggest that Fd59a serves as a transcription factor to regulate expression of genes involved in reproduction, cell growth and death in the testis.” Misexpression of these genes can be an indirect effect of intermediate genes. Do the authors claim that Fd59a regulate all misexpressed 1863 genes directly?
L362: serval potential Fox binding sites ïƒ several potential Fox binding sites
Author Response
1. Summary
Thank you very much for taking the time to review this manuscript. Please find the detailed responses below and the corresponding revisions/corrections highlighted/in track changes in the re-submitted files.
2. Point-by-point response to Comments and Suggestions for Authors
1. L74: “In Drosophila, expression of Fd59a/FoxD was about 2-fold higher in the testis than in the ovary.” Also
Answer:Revised as suggested.
2. L187: “Interestingly, microarray data from the Flybase showed that expression level 187 of Fd59a was about 2-fold higher in the testis than in the ovary.” However, microarray data available the Flybase show that from the Fd59a/FoxD has higher expression in ovaries than in testes: http://flybase.org/reports/FBgn0004896.htm. Please clarify.
Answer:In FlyAtlas Anatomical expression data, the expression levels of Fd59a in adult ovary and testis were 0.8 and 1.9, respectively, indicating that expression level of Fd59a was about 2-fold higher in the testis than in the ovary. We have changed “microarray data” to “FlyAtlas Anatomical expression data” in line 195.
3. L315: “Then, 2000 bp promoter sequences upstream the transcriptional start sites of these select genes were downloaded, and the potential Fox binding sites were predicted using JASPAR database (the relative profile score threshold was set to 85%).” These data are not available within the manuscript. Please provide these results.
Answer:The numbers of potential Fox binding sites of selected genes are provided in Table 3.
4. L321: “Taken together, our results suggest that Fd59a serves as a transcription factor to regulate expression of genes involved in reproduction, cell growth and death in the testis.” Misexpression of these genes can be an indirect effect of intermediate genes. Do the authors claim that Fd59a regulate all misexpressed 1863 genes directly?
Answer:We agree with you that Fd59a did not directly regulate all the misexpressed genes. Fd59a may directly or indirectly regulate these 1863 genes. “directly or indirectly” has been added to the end of the sentence (line 336).
5. L362: serval potential Fox binding sites à several potential Fox binding sites
Answer:Revised as suggested.
Reviewer 2 Report
Comments and Suggestions for Authors
In this study, Tang et al. describe the analysis of the Fd59a gene in the spermatogenesis of Drosophila. They found that mutations in Fd59a reduce male fertility, inducing spermatogenesis defects and increased apoptosis. By RNA-seq and RT-qPCR experiments, they also found that a number of genes appears differentially regulated in Fd59a mutants compared to the control.
The most of the the experiments have been performed using two mutant alleles analyzed in homozygous condition, in particular using the Fd59a2 null allele. When an homozygous mutation is analyzed, there is the possibility that the phenotypes of the mutant strain could derive from other mutations carried by the chromosome in homozygosis. Furthermore, both Fd59a1 and Fd59a2 derive from the same EMS-based genetic screen (Lacin et al., 2014 Developmental Biology). For this reason, the two strains have the same genetic background, and they may share the same unmapped mutations in the second chromosome. To exclude the possibility that the observed phenotypes derive from other mutations in the second chromosome, the authors need to produce hemizygous flies combining the Fd59a mutant alleles with a deletion encompassing the Fd59a locus. This is a common method used to confirm that a phenotype is specifically induced by a particular mutation. Alternatively, the authors need to complement the phenotype of the Fd59a mutant with a transgene carrying a wild type Fd59a gene.
Data presented in Figure 3A are ambiguous. Testis shown in Figure c2 is altered and very different from those shown in figure a2 and b2. Since the only morphological phenotype is the swelling of the anterior tip of the testis, the experiments need to be performed using samples that do not have alterations of unknown origin. Furthermore, enlargements shown in figures a3, b3, c3 were taken in different point along the anterior-posterior axis, and this may account for the observed differences. The authors need to perform precise quantitative measurements in the same region of the different testes analyzed. In Figure a2 the anterior tip of the testis is not visible and it is not possible to see in which point the enlargement has been taken.
While in Figure 3A both the Fd59a alleles are analyzed, in Figure 3B the data relative to the Fd59a1 allele are missing. Why the authors did not show this quantitative data?
In Figure 3C, in the male fertility test which females have been used? In Lacin et al., 2014 (Developmental Biology) it is reported that “crosses of wild-type females to fd59a mutant males produced normal numbers of eggs, but crosses of fd59a mutant females to wild-type males produced far fewer eggs (Fig. 9E). Crosses of fd59a mutant males and females produced even fewer eggs, suggesting that fd59a may play a subtle role in male mating or mating behavior that may impact egg laying when such males are crossed to fd59a, but not wild-type, females (Fig. 9E).”. The male fertility test have to be performed using control females also in the cross with the mutant males.
In Figure 4A, the resolution of the images is very low (in particular the red channel - a3-d3 and a5-d5), and it is difficult to distinguish between spectrosomes and fusomes. A higher resolution is required. This analysis has been made only at qualitative level. A quantitative analysis needs to be performed.
Figure 5A. The quality of these images is too low, in particular the morphology of testes shown in Figures a1-a3, b1-b3, c1-c3.
Figure 5B. Also in this case the morphology of the testes show in the figure is altered. It is necessary to performed this type of analysis in testes with a “normal” morphology. Furthermore, to compare differences in seminal vesicle, it is necessary to select the area to enlarge in the same position along the anterior-posterior axis.
By RNA-seq, the authors found that about 2000 genes are differentially regulated in Fd59a2/2 males. It is a very high number of genes, and it may be random that among these genes some play a role in gonad development and spermatogenesis. Furthermore, it is reported that the BDSC56820 strain used for these experiments caries at least one P{FRT} element in the second chromosome and two other mapped mutations. It is highly probable that these mutations have an effect on the transcriptome.
Data presented in Table 3 do not show the error nor a significance test. It is not reported how many biological replicates have been analyzed. Also in Material and Method the number of biological replicates is not reported. The validation of the RNA-seq data requires that RT-qPCR experiments are performed using replicates and doing a statistical analysis. Why the authors did not show these data using histograms? It would be easier to consult these data.
About the discussion, it is necessary to discuss the data of this study taking in consideration the work of Lacin et al. (2014).
Comments on the Quality of English LanguageThe English language of the manuscript is acceptable
Author Response
1.Summary
Thank you very much for taking the time to review this manuscript. Please find the detailed responses below and the corresponding revisions/corrections highlighted/in track changes in the re-submitted files.
2.Point-by-point response to Comments and Suggestions for Authors
1. Most of the experiments have been performed using two mutant alleles analyzed in homozygous condition, in particular using the Fd59a2 null allele. When a homozygous mutation is analyzed, there is the possibility that the phenotypes of the mutant strain could derive from other mutations carried by the chromosome in homozygosis. Furthermore, both Fd59a1 and Fd59a2 derive from the same EMS-based genetic screen (Lacin et al., 2014 Developmental Biology). For this reason, the two strains have the same genetic background, and they may share the same unmapped mutations in the second chromosome. To exclude the possibility that the observed phenotypes derive from other mutations in the second chromosome, the authors need to produce hemizygous flies combining the Fd59a mutant alleles with a deletion encompassing the Fd59a locus. This is a common method used to confirm that a phenotype is specifically induced by a particular mutation. Alternatively, the authors need to complement the phenotype of the Fd59a mutant with a transgene carrying a wild type Fd59a gene.
Answer:Thank you for the suggestion. We have crossed Fd59a2 flies with Df(2R) BSC864 flies which contain a deletion encompassing the Fd59a locus, and similar phenotypes of the testis were observed compared with Fd59a2/2 flies. The new results were included in revised Figure 3A (d1-d3, e1-e3).
2.Data presented in Figure 3A are ambiguous. Testis shown in Figure c2 is altered and very different from those shown in figure a2 and b2. Since the only morphological phenotype is the swelling of the anterior tip of the testis, the experiments need to be performed using samples that do not have alterations of unknown origin. Furthermore, enlargements shown in figures a3, b3, c3 were taken in different point along the anterior-posterior axis, and this may account for the observed differences. The authors need to perform precise quantitative measurements in the same region of the different testes analyzed. In Figure a2 the anterior tip of the testis is not visible and it is not possible to see in which point the enlargement has been taken.
Answer:We have replaced the original Figure 3A with a new Figure 3A. In the new Figure 3A, the apical regions of the testes from Fd59a mutant flies were swelling (Figure3A, b1, c1 and e1) and fewer mature sperms were observed in the seminal vesicles (Figure3A, b2-b3, c2-c3 and e2-e3) compared to the control flies (Figure 3A, a1-a3, d1-d3).
3. While in Figure 3A both the Fd59a alleles are analyzed, in Figure 3B the data relative to the Fd59a1 allele are missing. Why the authors did not show this quantitative data?
Answer:“As the phenotype of Fa59a2/2 flies was more significant than that of Fa59a1/1 flies, we carried out our following studies in the Fa59a2/2 mutant flies. (Lines 237-239)
4. In Figure 3C, in the male fertility test which females have been used? In Lacin et al., 2014 (Developmental Biology) it is reported that “crosses of wild-type females to fd59a mutant males produced normal numbers of eggs, but crosses of fd59a mutant females to wild-type males produced far fewer eggs (Fig. 9E). Crosses of fd59a mutant males and females produced even fewer eggs, suggesting that fd59a may play a subtle role in male mating or mating behavior that may impact egg laying when such males are crossed to fd59a, but not wild-type, females (Fig. 9E).”. The male fertility test have to be performed using control females also in the cross with the mutant males.
Answer:We used w1118 females to cross with Fa59a2/2 males for the male fertility test. (Lines 131-136, 239).
5. In Figure 4A, the resolution of the images is very low (in particular the red channel - a3-d3 and a5-d5), and it is difficult to distinguish between spectrosomes and fusomes. A higher resolution is required. This analysis has been made only at qualitative level. A quantitative analysis needs to be performed.
Answer:We have replaced the images with higher resolution ones as suggested for Figure 4A, and quantitative analysis of spectrosomes and fusomes has been performed and the results were included as new Figure 4B, C.
6. Figure 5A. The quality of these images is too low, in particular the morphology of testes shown in Figures a1-a3, b1-b3, c1-c3.
Answer: We have replaced the images with higher resolution ones as suggested for Figure 5A (a1-a4, b1-b4, c1-c4 and d1-d4).
7. Figure 5B. Also in this case the morphology of the testes show in the figure is altered. It is necessary to performed this type of analysis in testes with a “normal” morphology. Furthermore, to compare differences in seminal vesicle, it is necessary to select the area to enlarge in the same position along the anterior-posterior axis.
Answer:Corrections have been made and we have replaced the images with higher resolution ones as suggested for Figure 5B.
8. By RNA-seq, the authors found that about 2000 genes are differentially regulated in Fd59a2/2 males. It is a very high number of genes, and it may be random that among these genes some play a role in gonad development and spermatogenesis. Furthermore, it is reported that the BDSC56820 strain used for these experiments caries at least one P{FRT} element in the second chromosome and two other mapped mutations. It is highly probable that these mutations have an effect on the transcriptome.
Answer:We identified the differential expressed genes with at least 1.5-fold change (p-adjust < 0.05). In the select candidate genes related to spermatogenesis and cell death, several Fox binding sites were predicted in the promoter of each gene, while more than 10 potential Fox binding sites were in the promoters of Spd-2, Cal1, Blanks, Ptp52F, Lola and Debcl genes, suggesting that these genes are regulated by Fd59a directly. The mutations in the second chromosome may have an effect on the number of DEGs but did not influence the identification of candidate genes.
9. Data presented in Table 3 do not show the error nor a significance test. It is not reported how many biological replicates have been analyzed. Also in Material and Method the number of biological replicates is not reported. The validation of the RNA-seq data requires that RT-qPCR experiments are performed using replicates and doing a statistical analysis. Why the authors did not show these data using histograms? It would be easier to consult these data.
Answer: For each sample, three biological replicates and at least three technical replicates for each biological replicate were performed (lines 178- 184). In Figure 6C, we validated the RNA-seq data using qRT-PCR and the results were shown as histograms with statistical analysis. Data presented in Table 3 were from the RNA-seq data with p-value < 0.05.
10. About the discussion, it is necessary to discuss the data of this study taking in consideration the work of Lacin et al. (2014).
Answer: we have added discussions related to the work of Lacin et al in the discussion section (lines 350-351, 371-372).
Round 2
Reviewer 2 Report
Comments and Suggestions for Authors
After the revision of the manuscript, which took into account my requests, the manuscript is now suitable for publication.